# FLOT Versus CROSS—What Is the Optimal Therapeutic Approach for Locally Advanced Adenocarcinoma of the Esophagus and the Esophagogastric Junction?

**DOI:** 10.3390/cancers17152587

**Published:** 2025-08-06

**Authors:** Martin Leu, Hannes Mahler, Johanna Reinecke, Ute Margarethe König, Leif Hendrik Dröge, Manuel Guhlich, Benjamin Steuber, Marian Grade, Michael Ghadimi, Volker Ellenrieder, Stefan Rieken, Alexander Otto König

**Affiliations:** 1Department of Radiotherapy and Radiation Oncology, University Medical Center Göttingen, 37075 Gottingen, Germany; h.mahler@zahnarztpraxis-bohne.com (H.M.); hendrik.droege@med.uni-goettingen.de (L.H.D.); manuel.guhlich@med.uni-goettingen.de (M.G.); stefan.rieken@med.uni-goettingen.de (S.R.); 2Department of Gastroenterology, Gastrointestinal Oncology and Endocrinology, University Medical Center Göttingen, 37075 Gottingen, Germany; johanna.reinecke@med.uni-goettingen.de (J.R.); ute.koenig@med.uni-goettingen.de (U.M.K.); benjamin.steuber@med.uni-goettingen.de (B.S.); volker.ellenrieder@med.uni-goettingen.de (V.E.); alexander.koenig@med.uni-goettingen.de (A.O.K.); 3Department of General, Visceral, and Pediatric Surgery, University Medical Center Göttingen, 37075 Gottingen, Germany; marian.grade@med.uni-goettingen.de (M.G.); mghadimi@med.uni-goettingen.de (M.G.)

**Keywords:** FLOT, CROSS, esophageal adenocarcinoma, esophagogastric junction adenocarcinoma, locally advanced tumor

## Abstract

Patients with locally advanced adenocarcinoma of the esophagus (EAC) or the esophagogastric junction (AEGJ) are commonly treated with either perioperative chemotherapy using the FLOT regimen or neoadjuvant chemoradiotherapy following the CROSS protocol. This retrospective study evaluated 76 patients treated with one of these two approaches at a tertiary cancer center between 2015 and 2023. Our findings show that both strategies result in similar outcomes regarding survival, disease control, and surgical success. Treatment-related toxicities were comparable; however, patient compliance was higher in the CROSS group, where all patients completed the full radiotherapy regimen. These results suggest that neoadjuvant chemoradiotherapy is a well-tolerated and effective alternative to perioperative chemotherapy, showing comparable outcome trends in appropriately selected patients. This study contributes to the ongoing discussion about optimal treatment strategies for EAC and AEGJ and may help guide personalized therapy decisions.

## 1. Introduction

Adenocarcinoma of the esophagus (EAC) and the esophagogastric junction (AEGJ) represent rapidly increasing tumor entities in the Western world, especially among older individuals [1]. Despite ongoing developments in diagnostics and therapy, adenocarcinoma (AC) of the gastroesophageal junction (AEGJ) remains one of the major challenges in gastrointestinal oncology, associated with high morbidity and mortality rates. Although modern therapeutic regimens can increase the prognosis for patients with AEGJ and EAC, treatment-associated toxicity and patient-related comorbidities make it difficult to select suitable patients for the appropriate treatment regimen. Even in potentially curable tumor stages, the likelihood of locoregional recurrence or metachronous distant metastases influences the selection of neoadjuvant or perioperative treatment options, since both affect the prognosis and quality of life in patients with AEGJ and EAC cancer [2].

Radiochemotherapy has been established as neoadjuvant treatment option for patients with locally advanced EAC and AEGJ in addition to surgery for several decades and has been introduced into many national and international guidelines [3,4,5]. The CROSS trial included 366 patients and demonstrated a significant improvement in overall survival [6]. As an alternative to neoadjuvant chemoradiation (CRT), different protocols of perioperative (neoadjuvant and adjuvant) chemotherapy (ChT) have been established [7].

Two multicenter trials (MAGIC and ACCORD-07) have demonstrated a significant increase in overall survival with cisplatin-fluoropyrimidine-based perioperative chemotherapy plus surgery compared to surgery alone [8,9]. Additionally, oxaliplatin-based chemotherapy protocols (FLO/FLOT: 5-FU, leucovorin, oxaliplatin, +/− docetaxel) have been developed and demonstrated to be efficacious and safe for the perioperative treatment of EAC, AEGJ, and gastric cancer [7].

The multicenter, international, randomized phase III Neo-AEGIS study compared perioperative chemotherapeutic therapy (including ECF and FLOT) against neoadjuvant chemoradiotherapy as established in the CROSS trial in 377 patients with EAC and AEGJ. After three years, no notable differences were observed in terms of overall survival, perioperative morbidity, and quality of life [10]. However, the results have been discussed intensively, and interpretation remains difficult since epirubicine-containing regimes (ECF or EOX) turned out to be less effective than FLOT in patients with adenocarcinoma of the upper gastrointestinal tract. This incongruence in the chemotherapy arm complicates the comparison of neoadjuvant CROSS and perioperative chemotherapy in terms of efficacy and survival. There is only one randomized controlled trial (RCT), the German ESOPEC study, which compared neoadjuvant chemoradiotherapy (CROSS) with perioperative chemotherapy FLOT. The results were recently published with an overall survival benefit for the perioperative FLOT therapy [11].

Although randomized controlled trials such as ESOPEC provide high-level evidence, real-world data from tertiary centers complement these findings by reflecting treatment outcomes, feasibility, and adherence in unselected patient populations. This supports more personalized therapy decisions in routine clinical practice.

In the current study, we present information from our prospectively managed cancer database for locally advanced EAC and EGJ and compare the clinical outcomes and toxicity profiles of neoadjuvant chemoradiotherapy and perioperative chemotherapy.

## 2. Materials and Methods

### 2.1. Patients and Methods

We included 76 patients from prospectively maintained databases in the Comprehensive Cancer Center Göttingen (G-CCC) at the University Medical Center Göttingen (UMG) in Germany with adenocarcinoma of the esophagus or adenocarcinoma of the esophagogastric junction. Patients received either FLOT or CROSS treatment at our institution followed by surgical resection between January 2015 and March 2023. No other selection criteria were applied. Patients had extensive staging at the first contact with our institution which included clinical staging, imaging (CT or PET scan), esophagogastroduodenoscopy with endoscopic ultrasonography and biopsy, histopathological analysis, and blood tests. Tumor staging was based on the current guidelines of the German and European Cancer Societies [3,4]. The institutional ethical committee approved the data collection and scientific analysis (16/7/21).

Patients received either FLOT or CROSS treatment at our institution followed by surgical resection. The treatment allocation followed institutional guidelines: CROSS was generally preferred for Siewert I tumors, while FLOT was used for Siewert II and III cases.

### 2.2. Treatment and Follow-Up Strategy

Patients were treated in accordance with current German and European guidelines [3,4]. Multidisciplinary tumor boards evaluated all the patients before starting any treatment. The patients were followed-up at 3- to 6-month intervals at our institution for up to 5 years during or after treatment. In the radiotherapy department, the patients were assessed on a standardized basis at 18-month intervals up to 54 months during the follow-up period. Furthermore, a more frequent follow-up was conducted by the responsible gastroenterologist or surgeon. We used the Common Terminology Criteria for Adverse Events (CTCAE) version 5 to define acute oncology-related toxic complications [12].

### 2.3. CROSS

The RChT procedures were partially delineated by previous authors [13]. The patients were treated with photons of 6 MV and/or 20 MV. A CT planning scan was acquired with a slice thickness of at least 5 mm. The standard treatment regimen for patients was 41.4 Gy in 1.8 Gy fractions. If 3D conformal radiotherapy (3DCRT) was applied, an anterior–posterior field configuration was utilized. If required, supplementary lateral or oblique fields were incorporated. Intensity modulated radiotherapy (IMRT) and volume modulated radiotherapy (VMAT) were employed in 16 cases. Concomitant chemotherapy was administered weekly with the carboplatin area under the curve (AUC) 2 mg/mL/min and paclitaxel 50 mg/m^2^. In general, surgery was scheduled between four and six weeks after the administration of the final radiation fraction.

### 2.4. FLOT

FLOT was administered for four preoperative cycles, followed by four postoperative cycles which started 6 to 12 weeks after surgery. Each two-week cycle of FLOT comprised the administration of docetaxel 50 mg/m^2^ on day 1, oxaliplatin 85 mg/m^2^ on day 1, leucovorin 200 mg/m^2^ on day 1, and 5-FU 2600 mg/m^2^ as a 24 h infusion on day 1. Dose reduction was performed according to the physician’s choice as a result of toxicity or comorbidity. Surgery was scheduled four weeks after the administration of the final cycle.

### 2.5. Surgery

Prior to surgical intervention, the patients underwent comprehensive restaging, comprising diagnostic endoscopy and thoracic and abdominal CT imaging. All the patients were evaluated by an experienced surgeon and underwent further assessment by the multidisciplinary tumor board. Surgery was performed in accordance with the prevailing contemporary guidelines. Pathological assessment of tumor regression was conducted in accordance with the methodology proposed by Mandard et al. [14]. The Clavien-Dindo classification was used for postoperative complications [15].

### 2.6. Endpoints

The date of the histopathological diagnosis of the tumor was taken as the point of origin for the calculation of survival times. The primary endpoint for overall survival (OS) was death from any cause. Progression-free survival (PFS) was defined as the time elapsed until locoregional or distant tumor progression or death from any cause. Local and regional recurrences were regarded as endpoints for locoregional control (LRC). In the case of distant control (DC), the occurrence of distant metastasis was considered to be the endpoint.

### 2.7. Statistics

A comparison of patient and disease characteristics as well as toxicities was conducted using the Chi-squared and Kruskal–Wallis tests. The Kaplan–Meier estimator was employed for the purpose of evaluating survival statistics. The log-rank test was employed for the purpose of comparing survival times. In order to assess the potential influence of variables on survival, univariable Cox regression was conducted. Furthermore, variables that were found to be significant in the univariable analysis were also tested in a multivariable fashion. The software used was SPSS (version 29) and R (version 4.0.4), with the KMWin plugin [16]. A *p*-value of less than 0.05 was considered statistically significant.

## 3. Results

### 3.1. Patient Baseline, Radiochemotherapy, Perioperative Chemotherapy, Surgical, and Histopathological Characteristics

A total of 76 patients met the eligibility criteria for inclusion in the study. All the patients were treated over the course of a period spanning from January 2015 to March 2023. Thirty-six patients (47.4%) were treated in accordance with the FLOT regime, while 40 patients (52.6%) were treated in accordance with the CROSS protocol. The cohort comprised 64 male (84.2%) and 12 female patients (15.8%). The median age of the cohort was 64.0 years (range, 31–79 years). Due to patient mortality or loss to follow-up, the present study had a median follow-up period of 27.5 months (range, 2–101 months). A total of 75 patients underwent an R0 resection, with only one patient in the CROSS group undergoing an R1 resection. Please refer to Table 1 for a detailed overview of the distribution of the patient cohort according to the administered regimen. The patient groups differed statistically significantly in terms of initial T category, with a greater prevalence of T4 tumors in the FLOT group (FLOT n = 4 vs. CROSS n = 0, *p* = 0.02) and tumor localization, with a higher rate of AEG 1 tumors in the CROSS group (FLOT n = 10 vs. CROSS n = 26, *p* < 0.001) and a greater frequency of AEG III tumors in the FLOT group (FLOT n = 13 vs. CROSS n = 1, *p* < 0.001). No statistically significant difference was observed in the Charlson Comorbidity Index and ECOG. Adjuvant immunotherapy was administered to five of the forty patients in the CROSS group in accordance with the protocol of the Checkmate 577 trial (Reference). Of the 40 patients in the CROSS-group, 24 (60%) received 3D conformal radiotherapy, while 16 (40%) received intensity-modulated radiotherapy or volume-modulated arc therapy.

### 3.2. Toxicity, Treatment Compliance, and Surgical Outcomes

All 40 patients in the CROSS group received the full planned dose of 41.4 Gy. In the CROSS group, 7 out of 40 patients (17.5%) received less than five cycles of concomitant chemotherapy, whereas 18 out of 36 patients (50.0%) in the FLOT group received less than eight cycles of perioperative chemotherapy (*p* = 0.003) (shown in Table 2).

With regard to hematologic toxicity, no significant difference was observed in anemia ≥ III° (*p* = 0.13), leukopenia ≥ III° (*p* = 0.05), or thrombopenia ≥ III° (*p* = 0.59); however, no patient developed thrombopenia ≥ III°. A higher incidence of leukopenia was observed in the CROSS group compared to the FLOT group (*p* = 0.003), whereas a higher incidence of overall anemia was observed in the FLOT group (*p* < 0.001).

No patient died within 30 days of the surgical procedure. No statistically significant difference was observed in the median hospital stay following surgery between the two groups (*p* = 0.48). Four patients in the FLOT group and one patient in the CROSS group developed anastomotic leakage (*p* = 0.13). Regarding postoperative complications (Clavien-Dindo ≥ 3), we found no difference between the CRT (n = 3, 7.9%) and the ChT group (n = 6, 17.1%) (*p* = 0.31). We observed no significant difference in the resection status between the two groups, with only one patient presenting with microscopic residual disease (R1) in the CROSS group. Furthermore, no difference was observed in the ypT (*p* = 0.36) or ypN (*p* = 0.67) status. No difference in pathological complete remission status (*p* = 0.59) could be found. Patients who received CROSS therapy showed a trend towards better tumor regression (*p* = 0.08). When comparing the incidence of peritoneal carcinomatosis during follow-up, no differences were observed between the groups (*p* = 0.09) (shown in Table 3)

### 3.3. Survival Outcomes

The 3-year overall survival (OS), progression-free survival (PFS), loco regional control (LRC) and distant control (DC) rates were 65.5%, 44.4%, 65.1%, and 52.9%, respectively, across the entire study cohort. The 5-year LRC, PFS, OS, and DC rates were 60.9%, 40.5%, 65.1%, and 48.3%, respectively. A total of 20 patients (26.3%) experienced a locoregional recurrence. In the present study, 33 out of 76 patients (43.4%) developed distant metastases. A comparison of the FLOT and CROSS treatments revealed that the three-year rates for LRC were 61.6% vs. 68.6%, PFS 42.3% vs. 47.0%, OS 64.5% vs. 68.3%, and DC 49.8 vs. 56.5%. The five-year rates for LRC, PFS, OS, and DC were 61.5% vs. 68.6%, 33.9% vs. 42.8%, 60.2% vs. 63.4%, and 42.1% vs. 56.5%, respectively. However, these findings did not reach statistical significance in the log-rank test (OS *p* = 0.91, PFS *p* = 0.82, LRC *p* = 0.81, DC *p* = 0.39) (shown in Figure 1).

### 3.4. Uni- and Multivariable Analysis for Factors Affecting OS, PFS, and LRC

The univariable analysis demonstrated that ypT stage (3.28, 95% CI 1.25–8.59; *p* = 0.02), ypN status (5.91, 95% CI 3.04–11.50; *p* < 0.001), grading (HR 2.09, 95% CI 1.06–4.09, *p* = 0.03) as well as pathological complete remission (HR 0.20, 95% CI 0.05–0.83; *p* = 0.03) were determinant factors for PFS (shown in Table 4). In the multivariable analysis, ypN (HR 5.42, 95% CI 2.70–10.87; *p* < 0.001) remained significant for PFS (shown in Table 5). With regard to OS, grading (HR 3.27, 95% CI 1.38–7.69; *p* = 0.007) and ypN status (HR 5.18, 95% CI 2.11–12.68; *p* < 0.001) demonstrated a significant impact in the univariable analysis. In the multivariable analysis, grading (HR 2.53, 95% CI 1.01–6.05; *p* = 0.04) and ypN status (HR 4.19, 95% CI 1.67–10.51; *p* = 0.002) remained statistically significant. When assessed in a univariable fashion, ypT (HR 4.22, 95% CI 1.41–12.68; *p* = 0.01) and ypN (HR 3.73, 95% CI 1.51–9.24; *p* = 0.004) demonstrated a significant impact on locoregional control. In the multivariable model ypT status (HR 3.19, 95% CI 1.03–9.89; *p* = 0.04) and ypN status (HR 2.76, 95% CI 1.09–7.00; *p* = 0.03) remained statistically significant.

## 4. Discussion

Both preoperative radiochemotherapy (CRT) and perioperative chemotherapy (ChT) have been shown to be superior to surgery alone in several randomized trials [6,7,8,17]. However, a direct comparison of these two options has long been lacking, and the recommended preoperative or perioperative treatment regimen has been the subject of considerable controversy. Therefore, we analyzed our prospectively established database to evaluate outcomes in a more homogeneous patient population at our tertiary cancer center, focusing on the two most effective treatment regimens, FLOT and CROSS.

The evolution of neoadjuvant radiochemotherapy and perioperative systemic therapeutic approaches occurred largely independently, resulting in the establishment of both procedures as potential treatment options in numerous guidelines [3,4,5]. In clinical practice, this situation frequently presents a challenge when selecting one of the two options and has led to the development of various institutional treatment standards.

In our institution, the selection criteria for either FLOT or CROSS are as follows: proximal adenocarcinoma of the esophagus or esophagogastric junction (EGJ) (Siewert I) are more likely to undergo neoadjuvant radiochemotherapy, whereas perioperative chemotherapy using FLOT is typically reserved for Siewert II and III tumors. This is reflected in the distribution of EGJ carcinomas in our study, with more Siewert I (n = 26, 65.0%) carcinomas in the CROSS group and a majority of Siewert III carcinomas in the FLOT group (n = 28, 72.2%). The only additional imbalance was a higher rate of T4 tumors in the FLOT group.

The median follow-up period was 27.5 months (range, 2–101 months). As the majority of tumor-related events occur within the first two years after diagnosis, this follow-up is deemed sufficient to draw meaningful conclusions [13,18]. The median age of our cohort was 64.0 years (range, 31–79 years), comparable with other retrospective analyses such as those of Elshar et al. [19] and randomized controlled trials (RCTs) like ESOPEC [11]. Regarding gender, more male patients (n = 64, 84.2%) were treated, which is consistent with other investigations (Hoeppner et al., 89.3%; Elshal et al., 82.0%) [11,19], reflecting the underlying pathophysiological factors in carcinogenesis.

Since the resection status is a major prognostic factor for locally advanced EGJ and EAC, we found no differences in the R1 resection status between CRT and ChT in our cohort, consistent with the majority of other trials [19]. However, the absolute number of R1 resections (n = 23, 26.0%) reported elsewhere was significantly higher than in our analysis (n = 1, 1.3%) [19]. The ESOPEC trial reported 5.2% R1 resections in the ChT group and 3.9% in the CROSS group, both higher than in our population [11].

Although the difference in pCR rates between the two groups did not reach statistical significance in our cohort, the numerically higher rate in the CROSS group (17.5% vs. 13.9%) suggests a trend toward enhanced tumor regression with neoadjuvant chemoradiotherapy. This is consistent with prior trials such as Neo-AEGIS, where chemoradiotherapy resulted in improved regression but not necessarily improved survival compared to perioperative chemotherapy. Patients treated with CROSS showed a trend toward better tumor regression (*p* = 0.08), without reaching statistical significance. This is congruent with results from the Neo-AEGIS trial, which also reported improved tumor regression status for CRT; interestingly, the ESOPEC trial did not observe increased tumor regression after CRT [10,11].

Regarding survival, our study showed no differences in overall survival (OS), progression-free survival (PFS), loco-regional control (LRC), and distant control (DC) between CRT and ChT. Elshaer et al. similarly reported no differences in OS and disease-free survival [11].

The multicenter, international, randomized phase III Neo-AEGIS trial compared perioperative chemotherapy against neoadjuvant RChT after CROSS in 377 patients with adenocarcinomas of the esophagus and gastro-esophageal junction [10]. After a three-year follow-up, no significant differences were observed in OS, perioperative morbidity, or quality of life, consistent with our results. However, ECF chemotherapy, which is less effective than FLOT, was predominantly used, with only a subgroup receiving the superior FLOT regimen [7].

Multivariable Cox regression analysis identified ypN status as the strongest prognostic factor for OS and PFS in our study, confirming the findings of Bachmann et al. [20]. Neo-AEGIS reported higher rates of odynophagia in patients treated with CRT compared to the ChT group, consistent with our results. For other treatment-related toxicities, the CRT group was less likely to experience higher acute toxicities, partially aligning with our findings, where we observed a higher rate of anemia in the FLOT group [10]. Systemic recurrence was more common (43.4% vs. 26.3%) than loco-regional failure, consistent with Elshar et al., where distant metastases outnumbered local recurrences [19]. Due to suboptimal distant control rates, the CheckMate 577 study evaluated consolidative immunotherapy with nivolumab in 792 patients without complete remission after neoadjuvant CRT followed by surgery [21]. The study demonstrated that consolidation with nivolumab significantly extended disease-free survival, increasing median disease-free survival from 11.4 to 22.4 months compared to a placebo. Following approval, five patients in our cohort received adjuvant immune consolidation with nivolumab after CROSS treatment, in line with the CheckMate 577 trial. While the number is too small for statistical analysis, this observation highlights the early clinical implementation of immunotherapy in real-world practice. The integration of checkpoint inhibitors—particularly for patients with residual disease—marks a promising development. Future prospective trials should assess the optimal timing, patient selection, and long-term impact of immunotherapy in both adjuvant and neoadjuvant settings.

In our study, CRT was completed in 87% of the patients, with all the patients receiving the full radiotherapy dose, whereas perioperative ChT was completed in only 50%. Similarly, in the ESOPEC trial, only 52.5% of the patients completed adjuvant treatment, mostly due to treatment-related side effects, although the majority had ECOG 0 (72.6%) [11]. In our real-world data, the FLOT completion rate was even lower (50%), potentially influenced by a higher proportion of patients with ECOG ≥ 1 (53.9%), which may have affected the survival outcomes.

Recently published ESOPEC trial results compared RChT following CROSS with perioperative ChT using the FLOT regimen [11]. Among 438 randomized patients, the FLOT cohort demonstrated significantly higher three- and five-year OS rates in both intention-to-treat and per-protocol analyses. Secondary endpoints of PFS and pathological complete remission (pCR) also favored FLOT. No subgroup benefited more from preoperative CRT than perioperative ChT. Initial subgroup analyses indicated that patients with advanced T and N stages derived greater benefit from perioperative FLOT than from neoadjuvant CRT. Thus, perioperative ChT with FLOT should be the preferred option in current clinical practice. Nevertheless, clinical decision making must also account for individual patient factors and institutional practice patterns. While ESOPEC provides a robust foundation for standard of care, our real-world data reflect actual clinical implementation, highlighting therapy completion rates, toxicity profiles, and outcomes in a less-selected patient population. These insights complement RCT data and underscore the importance of individualized treatment decisions beyond trial settings.

The limitations of this study include its retrospective design, which may introduce selection bias and limit control over confounding factors as well as the absence of propensity score matching to account for baseline imbalance. The relatively small sample size and single-center setting may also limit the generalizability of the results. Despite these limitations, this study provides relevant real-world data to inform clinical decision making. Future prospective studies are warranted to refine patient selection and further personalize treatment strategies. A larger, multi-institutional cohort analysis would help validate these observations in a broader and more diverse patient population. Such collaborative efforts could also account for the institutional variations in the treatment protocols and patient selection, thereby improving the external validity.

Nevertheless, patient selection and allocation remain crucial. For patients unsuitable for perioperative FLOT therapy due to frailty or critical comorbidities, CROSS combined with adjuvant consolidation immunotherapy remains a feasible and well-tolerated alternative.

## 5. Conclusions

In this retrospective study comparing perioperative FLOT chemotherapy and neoadjuvant chemoradiotherapy using the CROSS protocol in patients with locally advanced esophageal or esophagogastric junction adenocarcinoma, both treatment approaches demonstrated comparable outcomes in terms of overall survival, progression-free survival, and loco-regional and distant disease control. The toxicity profiles and surgical results were similarly balanced. Notably, treatment compliance was higher in the CROSS group, with fewer patients failing to complete the planned regimen.

These findings, while limited by the small sample size and single-center setting and the small number of patients beyond three years of follow-up, support the use of neoadjuvant chemoradiotherapy as a well-tolerated and effective alternative to perioperative chemotherapy.

For clinical practice, this highlights the importance of individualized treatment decisions based on patient characteristics, comorbidities, and expected treatment tolerance.

Future prospective studies are warranted to refine patient selection and further personalize treatment strategies in this setting.

## Figures and Tables

**Figure 1 cancers-17-02587-f001:**
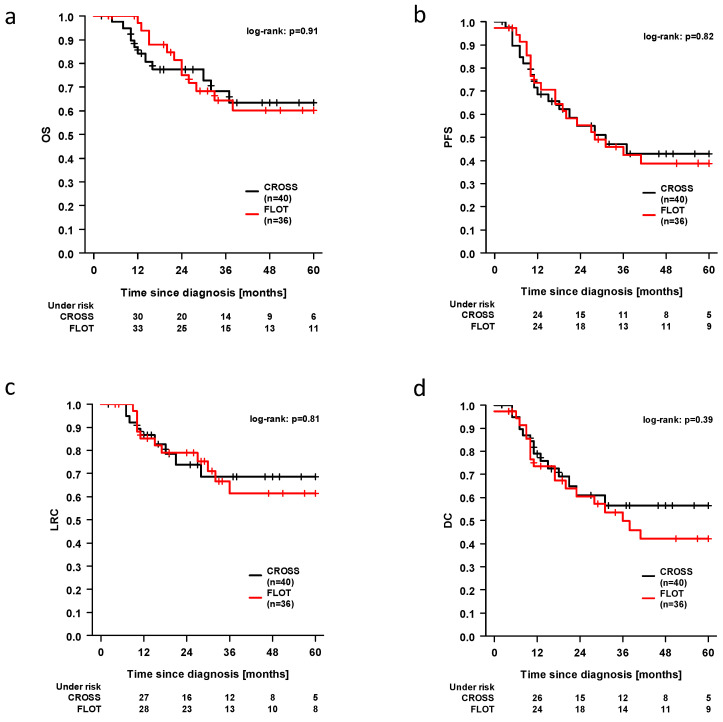
Kaplan–Meier plots for OS (panel (**a**)), PFS (**b**), LRC (**c**), and DC (**d**).

**Table 1 cancers-17-02587-t001:** Baseline patient, disease, and treatment characteristics.

Baseline Patient and Disease Characteristics			
Patient and Disease Characteristics, N (%)	FLOT (n = 36)	CROSS (n = 40)	*p*-Value *
Follow-up in months, median (min, max)	32.0 (4–90)	22 (2–101)	0.08
Age in years, median (min, max)	61 (38–79)	66 (31–77)	0.11
Female	6 (16.7)	6 (15.0)	0.92
Behavioral factors			
Smoking w/o regular alcohol	18 (50.0)	21 (52.5)	0.64
Smoking and alcohol abuse	3 (8.3)	2 (5.0)	0.60
Neither smoking nor regular alcohol	14 (38.9)	12 (30.0)	0.46
Undetermined	1 (2.8)	5 (12.5)	
BMI			0.33
<25	12 (33.3)	18 (45.0)	
≥25	24 (66.7)	22 (55.0)	
Tumor localization			<0.001
AEG I	10 (27.8)	26 (65.0)	
AEG II	13 (36.1)	13 (32.5)	
AEG III	13 (36.1)	1 (2.5)	
Initial T category clinical/ultrasound			0.02
cT1	1 (2.8)	2 (5.0)	
cT2	4 (11.1)	11 (27.5)	
cT3	27 (75.0)	27 (67.5)	
cT4	4 (11.1)	0	
Initial nodal status clinical/ultrasound			0.43
cN0	4 (10.1)	7 (16.7)	
cN+	32 (88.9)	33 (82.5)	
Radiotherapy			
Dose administered, median (min, max)	n.a.	41.4 (41.4–41.4)	
Dose planned, median (min, max)	n.a.	41.4 (41.4–41.4)	
Incomplete	n.a.	0	
Technique			
3D	n.a.	24 (60.0)	
IMRT/VMAT	n.a.	16 (40.0)	
CCI			0.11
0–3	13 (36.1)	6 (15.0)	
4–7	22 (61.1)	34 (85.0)	
8–10	1 (5.6)	0	
ECOG			0.07
0	14 (36.1)	22 (55.0)	
1	22 (61.1)	18 (45.0)	
2	2 (5.6)	0	
Adjuvant immunotherapy			
Yes	n.a.	5 (12.5)	

If not otherwise stated, per item the respective numbers with percentage in brackets are denoted * Chi-squared or Kruskal–Wallis test used; n.a. not available.

**Table 2 cancers-17-02587-t002:** Chemotherapy completion rates and incidence of acute hematologic toxicity.

Completion Rate and Acute Toxicity			
Hematologic Toxicity, N (%)	FLOT (n = 36)	CROSS (n = 40)	*p*-Value *
Anemia			
Anemia ≥ III°	4 (11.1)	1 (2.5)	0.13
0	0	13 (32.5)	<0.001
I°	18 (50.0)	24 (60.0)	
II°	14 (38.8)	2 (5.0)	
III°	4 (11.1)	1 (2.5)	
IV°	4 (11.1)	0	
Leukopenia			
Leukopenia ≥ III°	3 (7.9)	10 (25.0)	0.05
0	27 (75.0)	13 (32.5)	0.003
I°	2 (5.6)	9 (22.5)	
II°	4 (11.1)	8 (20.0)	
III°	3 (8.3)	10 (25.0)	
IV°	0	0	
Thrombopenia			
Thrombopenia ≥ III°	0	0	0.59
0	22 (61.1)	20 (50.0)	0.49
I°	13 (36.1)	17 (42.5)	
II°	1 (2.8)	3 (7.5)	
III°	0	0	
IV°	0	0	
Chemotherapy, N (%)			
Incomplete	18 (50.0)	7 (17.5)	0.003
Acute non-hematologic toxicity			
Fatigue ≥ II°	2 (5.6)	3 (7.5)	0.27
Nausea ≥ II°	3 (8.3)	2 (5.0)	0.67
Odynophagia ≥ II°	1 (2.8)	5 (12.5)	<0.001

For each parameter, the number of patients is provided, along with the percentage in brackets * Kruskal–Wallis test.

**Table 3 cancers-17-02587-t003:** Post-surgical tumor status and outcome.

Post-Surgical Tumor Status and Outcome			
Surgical Outcome, N (%)	FLOT (n = 36)	CROSS (n = 40)	*p*-Value *
Pathological complete response			
Yes	5 (13.9)	7 (17.5)	0.53
Tumor regression grade			0.08
1	13 (36.1)	20 (50.0)	
2	11 (30.6)	11 (27.5)	
3	9 (25.0)	4 (10.0)	
Undetermined	3 (8.3)	5 (12.5)	
Post-surgery T category			0.65
ypT0	8 (22.2)	7 (17.5)	
ypT1	2 (5.6)	5 (12.5)	
ypT2	5 (13.9)	5 (12.5)	
ypT3	13 (36.1)	21 (52.5)	
ypT4	7 (19.4)	0	
Undetermined	1 (2.8)	2 (5.0)	
Post-surgery nodal status ^+^			0.64
ypN0	21 (58.3)	24 (60.0)	
ypN+	15 (41.7)	14 (35.0)	
Pathological lymph node response ^+^			0.99
Yes	19 (52.8)	20 (50.0)	
No	17 (47.2)	18 (45.0)	
Resection status ^+^			0.33
R0	36 (100)	37 (92.5)	
R1	0	1 (2.5)	
Anastomotic leakage			
Yes	4 (10.0)	1 (2.6)	0.13
Death within 30 days post-surgery			1.00
Yes	0	0	
Hospital stay surgery			0.52
Median (min, max)	15 (9, 94)	14 (9, 76)	
Peritoneal carcinomatosis during follow-up			
Yes	7 (18.4)	2 (5.0)	0.10
Operation details			<0.001
Esophagostomy	3 (8.6)	35 (92.1)	
Extended Gastrectomy	32 (91.4)	3 (7.9)	
Clavien-Dindo			0.31
3 or higher	6 (17.1)	3 (7.9)	

For each parameter, if not otherwise specified, the number of patients and, in brackets, the percentage are given * Chi-squared test and Kruskal–Wallis test. ^+^ Data for two pat. in the CROSS-group are missing.

**Table 4 cancers-17-02587-t004:** Univariable Cox regression.

Variable	PFS		OS		LRC	
	Hazard Ratio	*p* Value	Hazard Ratio	*p* Value	Hazard Ratio	*p* Value
	(95% CI)		(95% CI)		(95% CI)	
Age (per year)	1.02 (0.99–1.05)	0.25	1.01 (0.96–1.05)	0.79	0.99 (0.95–1.03)	0.63
Sex						
Female (12) vs. male (64)	0.62 (0.24–1.58)	0.32	0.43 (0.10–1.82)	0.25	1.17 (0.39–3.52)	0.78
ECOG						
0 (34) vs. 1–2 (41)	0.89 (0.47–1.71)	0.73	0.64 (0.28–1.47)	0.29	0.49 (0.20–1.19)	0.12
BMI						
≥25 (46) vs. <25 (30)	0.54 (0.29–1.02)	0.06	0.72 (0.31–1.66)	0.44	0.44 (0.18–1.07)	0.07
Grading						
G3 (20) vs. G1–G2 (49)	2.13 (1.08–4.21)	0.03	3.27 (1.38–7.69)	0.007	1.46 (0.55–3.84)	0.45
Charlson Index						
>4 (36) vs. ≤4 (40)	1.01 (0.54–1.88)	0.97	1.02 (0.45–2.34)	0.95	0.41 (0.15–1.13)	0.09
Pathological complete response						
Yes (12) vs. no (62)	0.20 (0.05–0.83)	0.03	0.42 (0.10–1.78)	0.24	0.22 (0.03–1.61)	0.21
Post-surgery TNM						
ypT3–4 (41) vs. 1–2 (33)	2.31 (1.18–4.50)	0.01	1.30 (0.57–2.96)	0.54	4.22 (1.41–12.68)	0.01
ypN+ (29) vs. ypN0 (45)	5.65 (2.89–11.03)	<0.001	5.18 (2.11–12.68)	<0.001	3.73 (1.51–9.24)	0.004

The hazard ratios and 95% confidence intervals are given. *p* values < 0.05 were considered as statistically significant.

**Table 5 cancers-17-02587-t005:** Multivariable Cox regression analysis.

Variable	PFS		OS		LRC	
	Hazard Ratio	*p* Value	Hazard Ratio	*p* Value	Hazard Ratio	*p* Value
	(95% CI)		(95% CI)		(95% CI)	
Grading						
G3 (20) vs. G1–G2 (49)	1.51 (0.75–3.05)	0.25	2.53 (1.05–6.05)	0.04		n.a.
Pathological complete response						
Yes (12) vs. no (62)	0.53 (0.11–2.63)	0.44		n.a.		n.a.
Post-surgery TNM						
ypT3–4 (41) vs. 1–2 (33)	1.55 (0.73–3.33)	0.26		n.a.	3.19 (1.03–9.89)	0.04
ypN+ (29) vs. ypN0 (45)	5.42 (2.70–10.87)	<0.001	4.19 (1.67–10.51)	0.002	2.76 (1.01–7.01)	0.03

The hazard ratios and 95% confidence intervals are given; *p* values < 0.05 were considered as statistically significant; n.a. not available due to *p* > 0.05 in univariable analysis.

## Data Availability

The data that support the findings of this study are not publicly available due to privacy reasons but are available from the corresponding author upon reasonable request.

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
