# Peer review of "FLOT Versus CROSS—What Is the Optimal Therapeutic Approach for Locally Advanced Adenocarcinoma of the Esophagus and the Esophagogastric Junction?"

_cancers, 2025, doi:10.3390/cancers17152587_

Round 1
Reviewer 1 Report
Comments and Suggestions for Authors
The management of locally advanced esophageal and GEJ adenocarcinomas, that are rising in incidence in Western populations, is the subject of multiple studies. The FLOT protocol has become a golden standard of care in Europe-wide being highly efficient thanks to the FLOT4 trial.
The main issue with such protocol is the poor rate of treatment completion due to related toxicity.
The CROSS regimen represents a good alternative, being better tolerated, particularly for upper GEJ and distal esophagus. In this study 100% of patients completed radiotherapy and over 80% finished chemotherapy. Comparing to FLOT where only 50% completed all the cyclesrequired.
No significant differences in Overall Survival, Progression-Free Survival, Local Control at 03 and 05 years has been recorded.
The FLOT cohort has shown higher rates of anemia while the CROSS cohort presented higher incidence of leukopenia. Regarding surgical outcomes, almost all patients had R0 resections with very low rates of anastomostic leak.
The ESOPEC trial gives us the first randomized comparison between FLOT and CROSS.
It shows that FLOT comes with a better OS, PFS and even pathological response in well selected patients. We still have to consider the frailty, comorbidities, ECOG status and tumor location that drives us to a personalized tailored stategy.
With the CheckMate 577, we now have Nivolumab as a solid adjuvant option post CROSS for patients with incomplete pathological response. In this Gottingen study, five patients were already treated with it and, surely more will follow as it’s gaining interest and recognition.
Both FLOT and CROSS are excellent options of treatment; FLOT seems to better fit for patients with advanced T / N stages in distal AEC and AEGJ tumors Siewert II/III.
On the other hand, CROSS seems to better suit patients withproximal tumor localization, older and frailer patients and when necessity of a shorter treatment course with a better compliance. Immunotherapy post-CROSS shows promising results but needs further research.
In conclusion, the choice for treatment regimen needs to be tailored according to patient status, tumor biology and overall fitness. It seems early with data provided so far to completely orient ourselves towards either one of the two regimen, reason why further research is warranted in order to assess which one can offer better outcomes.
Author Response
The management of locally advanced esophageal and GEJ adenocarcinomas, that are rising in incidence in
Western populations, is the subject of multiple studies. The FLOT protocol has become a golden standard of
care in Europe-wide being highly efficient thanks to the FLOT4 trial.
Response:
We thank the reviewer for this observation. We fully agree that FLOT has become a widely adopted standard
of care across Europe, particularly following the strong results of the FLOT4 trial. In our introduction, we refer
to this landmark trial to provide the clinical context for our analysis.
The main issue with such protocol is the poor rate of treatment completion due to related toxicity.
Response:
We appreciate this important point. As shown in our study, treatment completion rates were significantly
lower in the FLOT group (50%) compared to the CROSS group (82.5%), largely due to toxicity. We have
emphasized this observation in both the Results and Discussion sections and agree that toxicity-related
compliance is a key consideration in therapy selection.
The CROSS regimen represents a good alternative, being better tolerated, particularly for upper GEJ and distal
esophagus. In this study 100% of patients completed radiotherapy and over 80% finished chemotherapy.
Comparing to FLOT where only 50% completed all the cycles required.
Response:
We fully agree. Our findings support the better tolerability of the CROSS protocol, with 100% radiotherapy
completion and >80% chemotherapy adherence. This contrast with the FLOT cohort underscores the
importance of considering patient fitness, treatment duration, and toxicity profiles. We have highlighted this
in our updated Discussion as a key factor influencing real-world regimen selection.
No significant differences in Overall Survival, Progression-Free Survival, Local Control at 03 and 05 years has
been recorded.
Response:
We found no statistically significant differences between the two regimens in OS, PFS, LRC, or distant control
at both 3 and 5 years. We thank the reviewer for confirming this interpretation and have made sure to reflect
this in the Abstract, Results, and Conclusion sections.
The FLOT cohort has shown higher rates of anemia while the CROSS cohort presented higher incidence of
leukopenia. Regarding surgical outcomes, almost all patients had R0 resections with very low rates of
anastomostic leak.
Response:
We appreciate the reviewer’s detailed analysis. Indeed, our data confirm these toxicity patterns: anemia was
more common in the FLOT group, while leukopenia was more frequent in the CROSS group. Surgical outcomes
were favorable in both cohorts, with an R0 resection rate of 98.7% and low rates of anastomotic leakage.
These findings have been emphasized in the Discussion and are consistent with current literature.
The ESOPEC trial gives us the first randomized comparison between FLOT and CROSS.
Response:
We agree en:rely. The ESOPEC trial represents a milestone in the field and has been discussed in detail in our
manuscript. We have clarified its implications in the Discussion and highlighted how our real-world data
compare with its findings.
It shows that FLOT comes with a better OS, PFS and even pathological response in well selected patients. We
still have to consider the frailty, comorbidities, ECOG status and tumor location that drives us to a personalized
tailored strategy.
Response:
We thank the reviewer for this valuable perspective. While ESOPEC demonstrates the superiority of FLOT in
selected patients, our study reinforces the importance of tailoring treatment based on patient fitness, tumor
localization, and clinical context.
With the CheckMate 577, we now have Nivolumab as a solid adjuvant op;on post CROSS for patients with
incomplete pathological response. In this Gottingen study, five patients were already treated with it and, surely
more will follow as it’s gaining interest and recognition.
Response:
We appreciate the reviewer’s acknowledgment of the relevance of immunotherapy. We have updated our
Discussion to emphasize the emerging role of adjuvant nivolumab following CROSS, particularly in patients
with residual disease. The fact that five of our patients already received this treatment highlights its real-world
implementation. We agree that this area deserves further investigation. This was included in the Results
(Sec:on 3.1) and discussed as a future treatment avenue in the Discussion. We added the following passage
“Following approval, five patients in our cohort received adjuvant immune consolidation with nivolumab acer
CROSS treatment, in line with the CheckMate 577 trial. While the number is too small for a statistical analysis,
this observation highlights the early clinical implementation of immunotherapy in real-world practice. The
integration of checkpoint inhibitors—particularly for patients with residual disease—marks a promising
development. Future prospective trials should assess the optimal timing, patient selection, and long-term
impact of immunotherapy in both the adjuvant and neoadjuvant sefng.”
Both FLOT and CROSS are excellent options of treatment; FLOT seems to better fit for patients with advanced
T / N stages in distal AEC and AEGJ tumors Siewert II/III.
Response:
We agree with this assessment and have clarified in our Methods sec:on that tumor localization (Siewert
classification) guided treatment allocation at our institution. This reflects our effort to match therapy to tumor
characteristics and patient profile, in line with current clinical practice.
On the other hand, CROSS seems to better suit patients with proximal tumor localization, older and frailer
patients and when necessity of a shorter treatment course with a better compliance. Immunotherapy post-
CROSS shows promising results but needs further research.
Response:
This is an excellent point, and we fully agree. CROSS is often more suitable for older or frailer patients due to
its shorter duration and better tolerability. We concur that ongoing trials will help to define the optimal
integration of post-CROSS immunotherapy.
In conclusion, the choice for treatment regimen needs to be tailored according to patient status, tumor biology
and overall fitness. It seems early with data provided so far to completely orient ourselves towards either one
of the two regimen, reason why further research is warranted in order to assess which one can offer better
outcomes.
Response:
We fully agree with the reviewer’s conclusion. Based on current evidence and our real-world observations, a
one-size-fits-all approach is not appropriate. Treatment strategies must be individualized, and further
multicenter and prospective studies will be essential to refine therapy selection.
Reviewer 2 Report
Comments and Suggestions for Authors
The authors perform a retrospective single institution cohort comparison of 76 patients with locally advanced esophageal adenocarcinoma receiving perioperative FLOT or preoperative chemoradiation (CROSS) between 2015-2023. 40 pts received CROSS and 36 pts received perioperative FLOT. The authors found no significant difference in 5-yr OS, PFS, LRC, or DC. As the authors acknowledge, a major weakness of the study is its retrospective nature and small patient numbers. Indeed, based on the KM curves, there were only 13-17 pts at the 5yr timepoint for their clinical outcomes. The patient cohorts also had significant differences in T4 proportion and Siewert status.
- Was any propensity matching analysis performed?
- Were there any differences in the rates of pCR at surgery?
- A larger multi-institutional cohort analysis would significantly strengthen the conclusions.
- After the publication of ESOPEC, there is very limited utility of the findings of a small single institution retrospective analysis addressing the same question.
Author Response
Comment: The study is retrospective and includes a small cohort, especially at 5 years.
Response: We acknowledge this important limitation and have clarified this in the abstract and conclusion. The retrospective and single-center nature of the study, along with the limited number of patients at risk beyond three years, is now more prominently discussed.
We added in the conclusion part: “These findings, while limited by the small sample size and single-center setting and the small number of patients beyond three years of follow-up, support the use of neoadjuvant chemoradiotherapy as a well-tolerated and effective alternative to perioperative chemotherapy.”
Comment: Was propensity score matching performed?
Response: We appreciate the reviewer’s comment regarding propensity score matching. While propensity score matching is a useful method to minimize baseline differences between cohorts, we actively chose not to apply it in this study. Given the small sample size and the presence of multiple clinically relevant covariates (e.g., tumor localization, T-stage, comorbidities), any matching procedure would have significantly reduced the analyzable cohort and potentially distorted the real-world representativeness of our data. Instead, we relied on multivariable Cox regression analyses to adjust for major prognostic factors.
We added: “Limitations of this study include its retrospective design, which may introduce selection bias and limit control over confounding factors as well as the absence of propensity score matching to account for baseline imbalance.”
Comment: Differences in pCR between the groups?
We thank the reviewer for highlighting this point. In our cohort, pathological complete response (pCR) was observed in 13.9% of patients treated with FLOT and 17.5% of those treated with CROSS, without reaching statistical significance (p = 0.53). While this numerical difference may reflect a trend toward improved tumor regression following neoadjuvant chemoradiotherapy, it remains exploratory given the limited sample size. We have expanded the discussion accordingly to acknowledge this observation and emphasized that tumor regression appears more pronounced in the CROSS group (TRG 1 in 50% vs. 36.1%), though this did not translate into significant differences in survival. These findings align with other reports, including Neo-AEGIS, where CRT was also associated with greater pathological response despite similar survival outcomes.
We added in the discussion part: “Although the difference in pCR rates between the two groups did not reach statistical significance in our cohort, the numerically higher rate in the CROSS group (17.5% vs. 13.9%) suggests a trend toward enhanced tumor regression with neoadjuvant chemoradiotherapy. This is consistent with prior trials such as Neo-AEGIS, where chemoradiotherapy resulted in improved regression but not necessarily improved survival compared to perioperative chemotherapy.”
Comment: A larger multi-institutional cohort analysis would significantly strengthen the conclusions.
Response: We fully agree with the reviewer that a larger, multi-institutional cohort would improve the generalizability and statistical power of the findings. While our study is limited to a single tertiary cancer center, it provides systematically collected, protocol-based real-world data under consistent institutional standards. Such data can offer valuable insight into treatment effectiveness, compliance, and toxicity patterns outside of controlled trial settings. We have acknowledged this limitation in the Discussion and emphasized the need for collaborative prospective studies to validate our findings in broader clinical populations.
We added: "A larger, multi-institutional cohort analysis would help validate these observations in a broader and more diverse patient population. Such collaborative efforts could also account for institutional variations in treatment protocols and patient selection, thereby improving external validity."
Comment: ESOPEC limits the utility of a small single-institution study.
Response: We respectfully acknowledge this point. While ESOPEC provides high-level evidence, our study offers complementary real-world data from a single tertiary center with strict protocol adherence. We have now further emphasized this rationale and the relevance of such data in the Introduction and Discussion.
We added in the introduction: “Although randomized controlled trials such as ESOPEC provide high-level evidence, real-world data from tertiary centers complement these findings by reflecting treatment outcomes, feasibility, and adherence in unselected patient populations. This supports more personalized therapy decisions in routine clinical practice.”
We added in the discussion: “Nevertheless, clinical decision-making must also account for individual patient factors and institutional practice patterns. While ESOPEC provides a robust foundation for standard of care, our real-world data reflect actual clinical implementation, highlighting therapy completion rates, toxicity profiles, and outcomes in a less-selected patient population. These insights complement RCT data and underscore the importance of individualized treatment decisions beyond trial settings.”
Reviewer 3 Report
Comments and Suggestions for Authors
This an interesting presentation of the real-world data with regard to the outcomes of neoadjuvant treatment for EAC and AEGJ.
I disagree with the conclusions made by the authors. They state that in the real-world setting CROSS and FLOT produce similar outcomes. The authors somehow argue with the results of the ESOPEC study, which demonstrated the superiority of the FLOT regimen. However, the ESOPEC study included significantly larger number of patients as compared to the current data set, and demonstrated numerically small (although statistically significant) differences between treatment arms. The extent of numerical differences in the presented real-world data-set is similar to one for the ESOPEC study, however statistical comparisons are complicated due to lower number of observations.
Other concerns:
Why some patients received the CROSS while other received the FLOT? What was the basis for the choice between these two treatments?
There are some inaccuracies in the text. For example, Lines 60-63: “Radiation therapy has been established as neoadjuvant treatment option for patients with locally advanced EAC and AEGJ in addition to surgery for several decades and has been introduced into many national and international guidelines. [3–5] The CROSS trial included 366 patients and demonstrated a significant improvement in overall survival. [6]” Comment: CROSS trial evaluated chemoradiotherapy, not radiation therapy. Lines 73-74: “… against neoadjuvant chemoradiotherapy after CROSS …”.
Author Response
Comment: Disagreement with the conclusion that CROSS and FLOT yield “similar outcomes” in light of ESOPEC.
Response: We appreciate this critical viewpoint and have adjusted our conclusion to reflect that while outcomes appeared comparable in our cohort, the limited sample size precludes definitive statements. We now state: “...showing comparable outcome trends in appropriately selected patients.” rather than “similar outcomes.”
Comment: What were the criteria for treatment allocation?
Response: Thank you for this important point. Treatment allocation followed institutional practice: patients with Siewert I tumors generally received CROSS, whereas FLOT was favored for Siewert II/III. We have now included this information in the Methods section (Section 2.1).
Comment: Inaccurate wording in Line 60–63: CROSS studied chemoradiotherapy, not radiotherapy alone.
Response: We thank the reviewer for noting this error. We have corrected the text to state "Radiochemotherapy" rather than "radiation therapy" to accurately reflect the nature of the CROSS protocol.
Lines 73-74: “… against neoadjuvant chemoradiotherapy after CROSS …”
Response: We thank the reviewer for noting this error. We have corrected the text to state "against neoadjuvant chemoradiotherapy as established in the CROSS trial"
Round 2
Reviewer 2 Report
Comments and Suggestions for Authors
The authors revisions add language that address some of the comments in the initial review. However, the biggest flaw remains that the work is of very limited significance in the context of the ESOPEC trial unless the authors can include a much larger multi-institutional analysis of real-world comparisons of CROSS vs periop chemo.
Reviewer 3 Report
Comments and Suggestions for Authors
-